# Fibroblast Growth Factor 23 to Alpha-Klotho Index Correlates with Systemic Sclerosis Activity: A Proposal for Novel Disease Activity Marker

**DOI:** 10.3390/jcm7120558

**Published:** 2018-12-17

**Authors:** Przemyslaw J. Kotyla, Aneta Kruszec-Zytniewska, Aleksander J. Owczarek, Magdalena Olszanecka-Glinianowicz, Jerzy Chudek

**Affiliations:** 1Department of Internal Medicine Rheumatology and Clinical Immunology Medical Faculty in Katowice Medical University of Silesia Katowice, 40-635 Katowice, Poland; 2Szpital Miejski w Sosnowcu, 41-200 Sosnowiec, Poland; aneta.zytniewska@gmail.com; 3Department of Statistics, Department of Instrumental Analysis, School of Pharmacy with the Division of Laboratory Medicine in Sosnowiec, Medical University of Silesia, Katowice, 41-200 Sosnowiec, Poland; aowczarek@paintbox.com.pl; 4Department of Pathophysiology, Medical Faculty in Katowice, Medical University of Silesia in Katowice, 40-752 Katowice, Poland; magolsza@gmail.com; 5Department of Internal Medicine and Oncological Chemotherapy, Medical Faculty in Katowice, Medical University of Silesia in Katowice, 40-038 Katowice, Poland; chj@poczta.fm

**Keywords:** fibroblast growth factor 23, α-Klotho, vitamin D, systemic sclerosis, disease activity

## Abstract

Systemic sclerosis, a connective tissue disease, is characterized by thickening of the skin, massive fibrosis of internal organs, vasculopathy, and immune system functioning aberration. Recently, vitamin D (VD) deficit, seen almost universally in patients with systemic sclerosis (SSc), has gained much attention. VD metabolism is precisely orchestrated at the level of the kidney by regulators: parathyroid hormone (PTH) and fibroblast growth factor 23 (FGF23) and their receptors with a FGF23 co-receptor—α-Klotho. The aim of this study was to assess the levels of VD, α-Klotho, FGF23 in SSc patients and to find the relationship between those parameters and disease activity. We enrolled 48 SSc patients with a diffuse variant of SSc and 23 sex- and age-matched healthy volunteers that served as the control group (CG). Patients were characterized by lower level of VD in comparison to CG (19.8 (12.6–28.9) vs. 24.5 (21.3–31.5) ng/mL; *p* < 0.01), significantly reduced levels of iFGF23 (19.3 (12.1–30.5) vs. 73.9 (59.7–110.2) pg/mL *p* < 0.001), and similar α-Klotho concentrations (1415 ± 557 vs. 1526 ± 397 pg/mL), respective. None of these parameters correlated with the extent of skin involvement (modified Rodnan Skin Score) and disease activity according to Eustar 2017 guidelines. The FGF23/α-Klotho index was significantly reduced in SSc patients (0.013 (0.0081–0.025) vs. 0.055 (0.038–0.095); *p* < 0.001), and its log_10_ correlated (*r* = 0.35; *p* < 0.001) with disease activity score (Eular2017). Our data showed that the FGF23/α-Klotho index may be considered as a novel, potential marker of systemic sclerosis activity.

## 1. Introduction

Systemic sclerosis (SSc) is a connective tissue disease of not completely understood pathogenesis characterized by autoimmune dysregulation, alterations in the microcirculation, and fibrosis of the skin and internal organs leading to their damage and subsequent insufficiency. According to the skin involvement distribution, SSc is categorized into limited and diffuse cutaneous SSc subsets (lcSSc and dcSSc, respectively) [1]. 

Vascular involvement is almost universal among patients with SSc and occurs early in the course of the disease. It presents initially as Raynaud syndrome that progresses further toward microangiopathy and vasculopathy causing chronic tissue ischemia and non-healing digital ulcers. At the moment there are no unique tools that may easily predict the disease course and outcome and more importantly characterize patients prone to rapid disease progression. Currently, patients’ clinical assessment relies on subjective measures as modified Rodnan Skin Score (mRSS), or requires expensive or invasive procedures (e.g., heart catheterization or high resolution computed tomography—HRCT). The diversity of the disease is also challenging as two distinct forms exist. Therefore, there is a need to discover useful, easily available, and cheap biomarkers that reflect fibrotic activity in the skin and internal organs as well as ongoing vasculopathy. To date, many potentially useful biomarkers have been proposed [2]. Most of them are useful in the prediction of the involvement of a specified organ but do not reflect disease activity in the whole body [3]. Quite recently, the European Scleroderma Trials and Research group (Eustar) proposed newly revised activity criteria for systemic sclerosis. The Eustar index included such variables as patient skin assessment, the extent of skin involvement measured by mRSS, tendon friction rubs, digital ulcers, (diffusion lung capacity for carbon monoxide—DLCO) value, and C-reactive protein concentration [4]. Unfortunately, Eustar index has not been tested against any currently available biomarkers yet. 

Recently, data has linked vitamin D (VD) deficiency to many autoimmune diseases, rheumatoid arthritis, systemic lupus erythematosus diabetes mellitus, multiple sclerosis, and systemic sclerosis [5,6,7,8,9]. The frequent findings of VD deficiency, particularly in people from developed countries might explain the high reported incidence of autoimmune disorders in these regions [6,10].

At the current level of knowledge, it is suspected that VD is a pleiotropic compound and its function goes far beyond calcium-phosphorus balance. Recent data suggested that VD is deeply involved into immune system functioning, being a potent modulator of innate and acquired immune response [11]. Among the many immunomodulatory mechanisms proposed to be involved in the interaction of VD with its receptor, the rapid-response steroid binding protein (1,25D_3_-MAARS) has attracted much attention. Activation of 1,25D_3_-MAARS directly modulates many signaling pathways including ones involved in the regulation of immune response such as cyclic guanosine monophosphate GMP and MapK kinases. Moreover, there is increasing evidence that VD regulates expression of many steps of signaling pathways including those activated by several cytokines, TNF, insulin, TGF B, and Wnt. Thus, any disturbance in VD concentration may lead to immune dysregulation and cause autoimmunity.

Vitamin D synthesis is precisely regulated at the level of the kidney. Two different proteins α-Klotho (FGF23 co-receptor) parathyroid hormone (PTH) and fibroblast growth factor 23 (FGF23) reduce VD activation (hydroxylation at the 1-α position) at the kidney level [12].

FGF23 is a 32kD protein synthesized mainly by osteocytes and osteoblasts and exerts its metabolic function by binding to its receptor–fibroblast growth factor receptor 1 (FGFR1) expressed at the distal tube. For sufficient receptor activation FGF23 requires the presence of its obligatory co-receptor Klotho [13]. The activation of FGFR1 inhibits the activity of 1-alpha hydroxylase, leading to the reduced synthesis of calcitriol and phosphaturia. The second major player in this field -α-Klotho a transmembrane and soluble protein with a long extracellular domain and a short cytoplasmic tail, was originally recognized as an anti-ageing protein, as the lack of *klotho* gene expression in mice leads to the accelerated ageing [14]. In line with this, Klotho deficiency is associated with tissue calcinosis, internal organ, and skin fibrosis as well as endothelial dysfunction [15]. The Klotho family consists of three single-pass transmembrane proteins, α-Klotho, β-Klotho, and γ-Klotho. Each of them combines with fibroblast growth factor receptors (FGFRs) to form receptor complexes for various FGFs. α-Klotho is a co-receptor for physiological FGF23 signaling and appears essential for FGF23-mediated regulation of mineral metabolism, including phosphate homeostasis. Patients with SSc are characterized by lower Klotho concentration [16,17,18,19]; however, the role of this finding remains obscure, as, in studies, Klotho was neither correlated with disease severity nor scleroderma-related damage. The aim of this study was to assess the simultaneous role of FGF23 and α-Klotho in the development of internal organ involvement in patients with the diffuse type of systemic sclerosis.

## 2. Material and Methods

Outpatients with systemic sclerosis were recruited from an observational cohort managed in the Department of Internal Medicine, Rheumatology, and Clinical Immunology. The diagnosis of the disease was established according to ACR/Eular 2013 criteria [20]. Patients were eligible for the study if they had the diffuse form of the disease according to LeRoy criteria [21]. 

### 2.1. Inclusion and Exclusion Criteria

Patients were excluded if they had any course of vitamin D supplementation in the last 6 months, overlapping syndrome SSc with rheumatoid arthritis or SSc with systemic lupus erythematosus (SLE), suffered from any malignancies or had changes in the immunosuppression therapy during the last 3 months. Pregnant and lactating women, as well as subjects with limited venous access, were also excluded from the study. Comprehensive medical history, including the time of formal SSc diagnosis, previous and concomitant treatment, and demographic data were taken in all patients, and detailed physical examination has been done according to the local guidelines. Routine laboratory tests including C_3_ and C_4_ complement levels and the antinuclear antibody profile (antinuclear and anti-centromere antibody on Hep2 cells, anti-Scl-70, Ro, La and Ro52 by enzyme-linked immunoabsorbent assay—ELISA) were retrieved from medical records. 

### 2.2. Imaging and Echocardiographic Studies

To identify patients with echocardiographic evidence of pulmonary hypertension, all patients underwent routine echocardiographic examination. Patients with estimated pulmonary arterial pressure in echocardiographic examination greater than 41 mm Hg were recognized as having elevated right ventricular systolic pressure (RVSP) on echo as recently proposed [22]. Functional lung assessment (FVC, TLC, and DLCO) was performed according to the American Thoracic Society/European Respiratory Society guidelines using standard equipment. Interstitial lung diseases were diagnosed based on high-resolution computed tomography CT scans (HRCT). Clinical manifestations of SSc included Raynaud’s phenomenon, skin damage (including pitting scars and active ulcers), and interstitial lung disease (defined as ground-glass opacities, sub-pleural reticulation with or without pleural irregularities, pleural traction, bronchiectasis, and/or honeycombing on HRCT scans). 

### 2.3. Concomitant Medications

Patients were on immunosuppressants (methotrexate < 25 mg/week; mycophenolate mofetil < 2.0 g/day; azathioprine < 200 mg/day, intravenous cyclophosphamide for interstitial lung diseases in dose not exceeding 1000 mg/month) and steroids (equal or less than 10 mg/day) when appropriate. In patients treated with cyclophosphamide, the assessment was performed at least 28 days after the last infusion to minimize the direct influence of cyclophosphamide on parameters studied in the laboratory analysis.

All demographic and disease-related data of patients are summarized in Table 1.

Detailed disease characteristics, including damage related to the disease as proposed by the Medsger and Eustar 2017 scale for disease severity and activity, were used [4,23]. Patient characteristics, as well as disease activity and damage assessment, were performed by the same researcher who was blinded to the results of α-Klotho, FGF23 and 25-hydroxyvitamin-D.

Twenty-three sex and age-matched apparently healthy subjects served as a control group.

The study was carried out according to the Declaration of Helsinki and the study protocol was approved by the local Bioethical Committee at the Medical University of Silesia, Poland (approval number KWN/0022/KB1/67/16). Patients and controls gave written informed consent prior to any study procedures.

Blood samples were taken in the morning after at least 8 hours of fasting. Blood samples were clotted for 30 min then centrifuged for 15 min at 1500 g. Samples were frozen immediately at −80 °C until laboratory assessment. The serum concentration of α-Klotho and intact FGF23 (iFGF23) were done using commercial ELISA kits (Immuno-Biological Laboratories Co., Ltd. Fujioka-Shi, Japan catalog number JP27998 and Immutopics, San Clemente, CA, USA catalog number 60-6600), total 25-hydroxyvitamin-D was analyzed on the same day with Cobas 422 analyses using electrochemiluminescence immunoassay with a detection range of 3–70 ng/mL (Roche Diagnostics, Manheim, Germany catalog number 06506780 160).

Antinuclear antibodies were assessed using ELISA kits produced by Generic Assay Dahlewitz Germany. Assays and tests were carried out according to the manufacturer’s instructions.

### 2.4. Data Analysis

Vitamin D status was classified based on predefined 25-OH-D levels as optimal >30 mg/dL, insufficient—between 10 mg/dL and 30 mg/dL and <10 mg/dL deficient as recently proposed [16].

### 2.5. Statistical Analysis

Statistical analysis was performed with Statistica 12.0 software (TIBCO Software Inc., Palo Alto, CA, USA). Nominal and ordinal data were expressed as percentages, while interval data were expressed as the mean value ± standard deviation in the case of normal distribution or as median with lower and upper quartile in the case of data with skewed or non-normal distribution. Distribution of variables was evaluated by the Shapiro–Wilk test and the quantile-quantile (Q-Q) plot, homogeneity of variances was assessed by the Fisher test. For comparison of data between the control and study group, the t-Student test for independent data (in the case of normal data distribution or after logarithmic normalization—if appropriate—in the case of skewed distribution) or the non-parametric U Mann-Whitney test (in non-normal data distribution) were used. The Pearson correlation coefficient was used as a measure of association between analyzed variables. The covariance analysis was used to assess the relationship between disease activity and vitamin D concentration. All tests were two-tailed. The results were considered as statistically significant with a *p*-value less than 0.05.

## 3. Results

We enrolled 48 patients aged 55 ± 19 years (30 women 18 men) with diffuse systemic sclerosis and 23 healthy controls aged 51 ± 12 years (15 women 8 men). Patients and controls did not differ with respect to age and sex distribution (Table 1). All patients were anti-nuclear antibody ANA positive, and antibodies were further identified as anti-topoisomerase I in 30 patients (62.5%), centromere in 12 patients (25%), and Ro-52 in 6 patients (12.5%). Patients presented with typical scleroderma-related symptoms, which are summarized in Table 1. The patients were treated with steroids, methotrexate, intravenous cyclophosphamide, mycophenolate mofetil, and azathioprine.

As compared to healthy subjects, patients with SSc were characterized by lower 25-OH-D concentration (Table 2). The optimal vitamin D level was noted only in 9 (18.8%), insufficiency in 32 (66.6%), and deficiency in 7 (14.6%) patients. No differences were observed when we analyzed 25-OH-D levels in patients with scleroderma-related complications: interstitial lung diseases and pulmonary arterial hypertension. The iFGF23 level was significantly lower in patients with SSc than in healthy counterparts (Table 2). 

In general, the concentration of iFGF23 did not differ between female and male patients 26.7 ± 18.3 vs. 24.9 ± 13.9 ng/dL and was independent of age. There was a weak positive correlation between log_10_ levels of Vitamin D and iFGF23 (*r* = 0.27; *p* < 0.05).

We did not observe any differences in α-Klotho level between patients and controls. In the study, α-Klotho levels did not correlate with disease duration, age, and gender. 

As the study was designed to assess the role of α-Klotho and iFGF23 in VD metabolism in SSc patients and their potential influence on disease severity and activity in SSc patients, correlation studies have been performed. In our group of patients, no associations were observed between vitamin D, α-Klotho level, and iFGF23, on the one hand, and the extent of skin involvement (mRSS value), disease severity (Medsger scale), and activity (Eustar2017 scale), on the other. Since α-Klotho and iFGF23 contribute mutually to the regulation of active VD levels we proposed a novel index (iFGF23/α-Klotho) that may reflect simultaneously changes in both proteins at the same time. We have found that in SSc patients log_10_ iFGF23/α-Klotho value was significantly lower in comparison to controls and this index significantly correlated with disease activity measured with Eustar2017 Scale (*r* = 0.35, *p* < 0.05) (Figure 1). Contrary to this, none of disease activity metrics alone correlated with FGF23/α-Klotho index that included mRSS total Medsger score, (delta skin- (worsening/improvement of skin status reported by patients), digital ulcers, tendon friction rubs, C- reactive protein, and the reduction of DLCO < 70% of the predicted value). To check the impact of VD concentration on FGF23 and α-Klotho level, we have divided the group of patients with SSc into two subgroups: with vitamin D levels below (*n* = 37) and over 30 ng/mL (*n* = 11). Based on this we have performed a covariance analysis with total Eustar values and checked if there is an influence of vitamin D concentration and disease activity on the log_10_ iFGF23/α-Klotho index. There was no statistically significant influence of vitamin D insufficiency/deficiency on the index (*p* = 0.97). However, a significant influence of total Eustar was observed (*p* < 0.01), but without statistically significant interaction between these two analyzed variables (*p* = 0.15). Based on this we may conclude, that the correlation between the log_10_ iFGF23/α-Klotho index and disease activity is independent to vitamin D concentration. All correlation analyses were presented in Table 3

## 4. Discussion

The role of VD in the pathogenesis of connective diseases is intensively debated and VD is currently recognized as a potent pleiotropic and immunomodulatory factor. The role of VD was recently addressed in the meta-analysis of An et al. who showed a reduction of VD in patients with SSc [24]. An et al. focused on patients from Italy, China, Brazil, and Israel—regions with relatively high sun skin exposure. In our study, only Caucasians living in Central Europe, a region with relatively low sun exposure, and common occurrence of VD deficiency, were enrolled. Therefore, VD deficiency is not only related to sun exposure. There are several explanations for the VD deficit burden in SSc patients that, in turn, translates to higher disease activity. The hardening of the skin may potentially contribute to reduced VD synthesis. The other explanation is reduced physical activity and indoor style of life commonly observed in patients that corresponds to limited sun ray exposure and reduced VD synthesis. Finally, the majority of SSc patients suffer from gastrointestinal tract involvement resulting in intestinal malabsorption that contributes to reduced vitamin D absorption

Less is known about α-Klotho and FGF23 in this regard. Contrary to published studies so far, we demonstrated that the Klotho level is unchanged in comparison to healthy subjects [16,18,19]. We, however, focused on α-Klotho since the previous studies measured the “total”, e.g., α and β-Klotho together, which may have contributed to the discrepancies observed between our study and studies already published. The VD is a potent inducer of α-Klotho expression in human proximal kidney cells; therefore, all states of VD deficiency may contribute to a reduction of the α-Klotho level. On the other hand, α-Klotho regulates negatively 25(OH)D_3_ -1-α-hydroxylase (encoded by CYP27B1 gene) resulting in the suppression of enzyme activity and reduction of bioactive VD level. Keeping in mind the role of α-Klotho as a potent suppressor of VD synthesis it may be speculated that the normal values of Klotho observed in our study mirrors the balance between reduced expression of α-Klotho in states of VD deficiency and the concomitant re-suppression of Klotho by reduced levels of iFGF23. It was recently described that FGF23 acts as a putative repressor of Klotho expression [25]. Therefore, in the settings of connective tissue diseases and perhaps also other inflammatory states, the α-Klotho level may be determined by the balance between reduced Klotho synthesis caused by VD deficiency and inflammation and direct activity of FGF23. We observed a significant reduction of FGF23, in our SSc patients, that may explain the observed balance between α-Klotho and FGF23. To our best knowledge, the reduction of FGF23 in SSc patients was reported in the literature for the first time. There are only two studies focused on FGF23 that have reported the FGF23 level unchanged in comparison to healthy counterparts. Contrary to these studies we observed a reduction of FGF23, the second major player in this model. In the study of Ahmadi et al. normal FGF23 level coincides with a reduction of α-Klotho that is the opposite of what was observed in our study, where the reduction of FGF23 and normal α-Klotho level have been shown [16]. The reason for the observed discrepancies is not clear. 

As with results from the literature, in our study any of parameters alone did not correlate with diseases activity, severity, and extent of skin involvement. Keeping in mind opposite expression of FGF23 and α-Klotho in inflammatory states, we proposed an iFGF23/α-Klotho index that may better reflect concomitant changes in α-Klotho and FGF concentration. To our surprise, for the first time, we demonstrated a relationship between α-Klotho and FGF and systemic sclerosis activity (measured in Eustar 2017 index). It is a new observation, because in previous studies a relationship between α-Klotho, iFGF23, and SSc activity was not observed. The observations from our study suggest an association between iFGF23/α-Klotho axis and the activity of SSc; however, this relationship is more complicated than previously suspected as it involves at least three elements: α-Klotho, VD, and FGF. Studies investigated the relationship between vitamin D level and disease outcome parameters gave contradictory results and the relation between low vitamin D and SSc still remains unclear [26]. Therefore, we suggest that iFGF/α-Klotho index is a potential marker of diseases activity that may reflect disease activity changes in SSc patients. This finding, however, should be tested in a larger study and more parameters should be considered, such as PTH, renal function, and cytokine profile.

### Limitation of the Study

Some limitations of the study should be addressed. Firstly, the study comprised 48 patients with systemic sclerosis, which is a large group but relatively small in relation to the size of the population with SSc; the relatively small group of subjects studied may have a strong impact on statistical analyses, final conclusions may therefore not be free of bias. We suggest that our preliminary observation should be verified in a larger, perhaps multicenter study which enables to broaden the final conclusion. Secondly, we recruited only patients with the diffuse form of the disease, so our conclusions are restricted solely to this population. Lastly, we did not perform an analysis of other factors involved in FGF23/α-Klotho/VD axis—namely parathormone, phosphaturia, and calcium levels. That is why we cannot exclude the participation of these factors in the regulation of the subtle balance between FGF23 and α-Klotho and VD in systemic sclerosis.

## Figures and Tables

**Figure 1 jcm-07-00558-f001:**
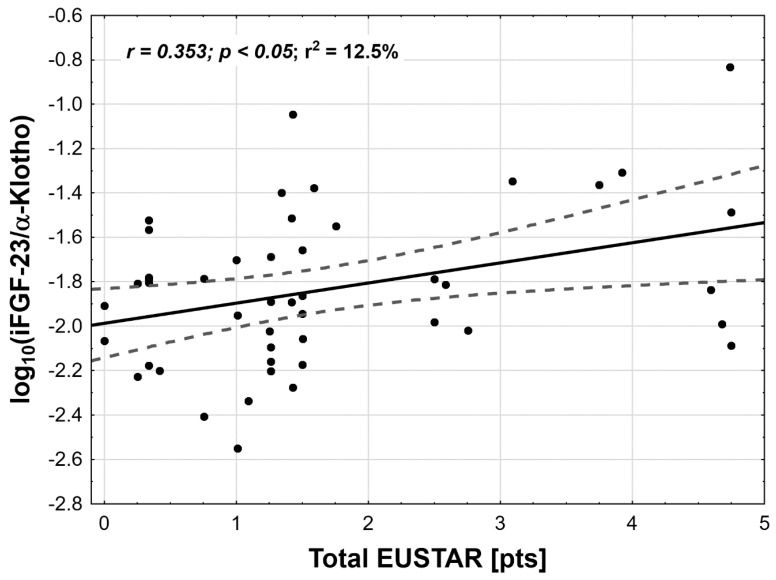
Correlation between the log_10_ FGF23/α-Klotho index and disease activity (Eustar 2017) in systemic sclerosis patients (FGF/α-Klotho values were presented with logarithmic normalization due to skewed data distribution).

**Table 1 jcm-07-00558-t001:** Clinical characteristics of patients with systemic sclerosis and control group.

Parameter	SSc Patients (*n* = 48)	Controls (*n* = 23)	*p*
Age (years)	55 ± 19	51 ± 12	*p* = 0.24
Sex (male/female)	18/30	8/15	*p* = 0.82
Disease duration years (range)	4 (2–7)		
Smoking status			
● Current	3 (6.3%)	3 (13%)	*p* < 0.01
● Ex-smoker	23 (47.9%)		
● never	22 (45.8)		
mRSS	11.2 ± 7.3		
Antibody profile			
● Anti-Scl-70	30 (62.5%)		
● Anti-centromere	12 (25%)		
● Anti-Ro-52	6 (12.5%)		
Treatment			
● Steroids	19 (40%)		
● Methotrexate	10 (20.8%)		
● Azathioprine	4 (8.3%)		
● Mycophenolate Mofetil	12 (25%)		
● Cyclophosphamide i.v.	18 (37.5%)		
Clinical presentation			
● Interstitial lung disease	40 (83.3%)		
● Raynaud	44 (91.7%)		
● Pulmonary hypertension	8 (16.7)		
● Finger ulcerations	12 (25%)		
● Arthritis	10 (20.8%)		
● Tenosynovitis	6 (12.5%)		
● Gastrointestinal tract dysmotility	4 (8.3%)		
● Heart rhythm disturbances	7 (14.6%)		
Laboratory data			
● Hemoglobin concentration (g/L)	12.91 ± 1.18		
● HCT (%)	38.27 ± 0.29		
● Creatinine (mg/dL)	0.78 ± 0.29		
● AST (IU)	16.0 (14.0–21.0)		
● ALT (IU)	15.0 (10.5–19.5)		
● GGTP (IU)	17.5 (1.5–26.1)		
● CRP (mg/dL)	5.0 (5.0–8.7)		
● ESR (mm/h)	27 ± 17		

Data presented as mean ± SD, range or median and lower; upper quartile where appropriate.

**Table 2 jcm-07-00558-t002:** Vitamin D, FGF 23, α-Klotho and FGF23/Klotho index in patients with SSc and healthy subjects.

Parameter	Patients with SSc (*n* = 48)	Healthy Controls (*n* = 23)	Significance
Vitamin D (ng/mL)	19.81 (12.59–28.94)	24.46 (21.28–31.48)	*p* < 0.01
iFGF 23 (pg/mL)	19.27 (12.13–30.52)	73.88 (59.66–110.2)	*p* < 0.001
α-Klotho (pg/mL)	1415.06 ± 556.76	1525.69 ± 397.30	*p* = 0.49
FGF23/α-Klotho index	0.013 (0.0081–0.025)	0.055 (0.038–0.095)	*p* < 0.001

Data presented as mean ± SD or median and lower–upper quartile where appropriate.

**Table 3 jcm-07-00558-t003:** The Pearson correlation coefficient and significance level between Vitamin D, FGF 23, Klotho and FGF23/Klotho index and disease activity scales (mRSS, Eustar, Medsger) in patients with SSc.

	Eustar (pts)	Medsger	mRSS
log_10_(iFGF-23 (pg/mL))	*r* = 0.23; *p* = 0.11	*r* = 0.02; *p* = 0.90	*r* = 0.09; *p* = 0.52
α-Klotho (pg/mL)	*r* = −0.14; *p* = 0.33	*r* = −0.21; *p* = 0.14	*r* = 0.10; *p* = 0.48
log_10_(VD (ng/mL))	*r* = 0.02; *p* = 0.92	*r* = −0.01; *p* = 0.97	*r* = −0.02; *p* = 0.92
log_10_(iFGF-23/α-Klotho)	*r* = 0.35; ***p* < 0.05**	*r* = 0.15; *p* = 0.29	*r* = 0.09; *p* = 0.56

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
