# Peer review of "Fibroblast Growth Factor 23 to Alpha-Klotho Index Correlates with Systemic Sclerosis Activity: A Proposal for Novel Disease Activity Marker"

_jcm, 2018, doi:10.3390/jcm7120558_

Reviewer 1 Report

1) There are many problems with grammar and sentence structure as well as some misspellings.  The manuscript needs to be more closely scrutinized for these issues, and they should be corrected prior to publication.

2) Page 2, line 47, the authors state: "The frequent findings of VD deficiency, particularly in people from well developed countries might explain the high reported incidence of autoimmune disorders in these regions."  To my knowledge a cause/effect relationship between low vitamin D and autoimmune disease has not been established.  If it has been please include the specific citation.  If not then I think this statement should be removed.

3) page 2, line 64, is "tube" supposed to be "tubule?"

4) page 2, lines 67-69, is this a reference to the klotho knockout mouse?  If so then the statement should add the qualifier "in mice" and cite the paper or an appropriate review describing this mouse. If the sentence refers to human data on klotho expression and aging please include the specific citation and be more specific in this sentence about what was found.

5) Citation 11 (Ahmadi et al.) reported lower Klotho levels and similar FGF-23 levels in SSc patients compared to controls, and citations 12 (Hajialilo et al.), 13 (Talotta et al.), and 14 (Talotta et al.) found lower Klotho levels in SSc vs controls.  The authors attribute the discrepancy in klotho levels to measurement of alpha-klotho levels (in this study) vs total klotho levels in the other studies.  Please comment on the distinct role of alpha-klotho compared to other subtypes, and why alpha-klotho was specifically measured in this study rather than all klotho.

6) Page 3, lines 115-116: please include the specific catalog numbers or other identifying information for each ELISA kit in case the companies sell more than one for that target protein.

7) Table 1: The mRSS distribution of 11.2 +/- 7.3 seems lower than expected for a cohort including only Diffuse patients.  This implies at least one patient had a score of only 4, correct?

8) Table 1: I would change "Pulmonary hypertension" to "Elevated RVSP on echo" or "echocardiographic evidence of Pulmonary hypertension," since echo parameters were used rather than right heart cath.  It is ok and understandable that right heart caths weren't done, but it is misleading to say without qualification that these patients had pulmonary hypertension based on echo alone.

9) It would have been better to measure and report the creatinine values of the healthy controls in addition to the SSc patients,  since klotho levels have an association with GFR.

10) Page 5, line 148: Don't say "We failed to show any differences..." Just say "no differences were observed."

11) page 5, lines 157-158: the p of 0.07 does not portray the strength or weakness of the correlation observed. Please provide the correlation co-efficient.

12) page 5, Table 2: I don't understand why ESR was included in this table.

13) page 5, lines 164-166: awkwardly worded. Can it just say "No associations were observed between …" and then list the variables that were analyzed?

14) page 6, figure 1: please explain why these data were log transformed

15) page 6, figure 1: It would be informative to perform separate analyses with adjustments for age, smoking status, GFR, ESR, and vitamin D levels, i.e., other variables that might be associated with FGF23 and/or Klotho levels. Could the association of low FGF-23 with SSc have been driven in part by the association between low Vitamin D and SSc?

16) Please provide more detail of the disease activity metrics.  Were there a few that were particularly associated with the FGF23/Klotho index?

17) page 6, lines 181-182: I can't tell what is meant by "VD deficiency seems to be mostly related to disease per se."

18) page 6, lines 184-186: Another explanation of low vitamin D in SSc is impaired GI absorption.

19) page 6-7, lines 203-215. I have a hard time understanding this discussion. In citation 11 the klotho levels were lower and FGF-23 levels similar in SSc.  Therefore the FGF-23/klotho index would have been in the opposite direction (higher nominator, lower denominator).  It might be better just to say that the reasons for the observed discrepancies are not yet clear.

20) page 7, line 121: I recommend not writing "it is a very important observation." Leave that to the readers' judgement.

21) line 122: please change "previous studies failed to show any relationship." An observation of no association is not a failure if the study was done well. Please say instead that "a relationship was not observed" or something that doesn't imply a judgment of success or failure depending on whether association was found.

22) page 7, line 222-223, the findings here provide evidence of an association between FGF-23/Klotho and SSc activity. I'm not sure what is meant by "a direct link."  The association might be in either direction or without any cause/effect relationship at all, and might be influenced by countless other factors.

23) page 7, lines 227-228: what is the evidence that iFGF/Klotho index is a marker of disease activity "in states of VD deficiency?"  Was the relationship only observed in SSc patients with low vitamin D?  The authors didn't report the associations in subgroups of patients with normal vs low vitamin D.

24) page 8, line 231: please be more specific about "statistical bias." Do you mean that the results could have been influenced by random chance due to low sample size, or, that there were biases?

25) In the methods section, please comment on whether the investigator measuring disease activity index was blinded to the lab variables (FGF-23, Klotho, vitamin D) under study.

26) I think for work of this nature, looking for biomarkers of SSc disease activity, it should be discussed in the background or discussion what the current state of knowledge is on biomarkers in SSc.  Are there any other biomarkers with associations with SSc disease activity? If so what are they, and how strong were the associations?

Author Response

We do really appreciate Reviewer attitude to improve our work. We did follow the suggestions and revised text subsequently.

Reviewer 1

There are many problems with grammar and sentence structure as well as some misspellings.  The manuscript needs to be more closely scrutinized for these issues, and they should be corrected prior to publication.

Some of grammatic errors have been corrected and English native speaker

Ad 1 The body of manuscript has been intensively checked for spelling and grammar errors and all necceasry corrections have been done by the professional English editing team provided by MDPI

2) Page 2, line 47, the authors state: "The frequent findings of VD deficiency, particularly in people from well developed countries might explain the high reported incidence of autoimmune disorders in these regions."  To my knowledge a cause/effect relationship between low vitamin D and autoimmune disease has not been established.  If it has been please include the specific citation.  If not then I think this statement should be removed.

Ad2

There are several data from recent studies that at least theoretically provide the link between risk of development of various form of autoimmunity  and low vitamin D status. This problem has been recently summarized by Danker  et al.Front Immunol. 2016; 7: 697. To better clarify the structure of manuscript  Danker et al has been cited again as well as the paper of   Lerner A, Jeremias P, Matthias T. The world incidence and prevalence of autoimmune diseases is increasing. Int J Celiac Dis (2015) 3(4):151–5.10. that reviewed the current data on increasing prevalence and incidence of autoimmunity

3) page 2, line 64, is "tube" supposed to be "tubule?"

Ad3. Correct

4) page 2, lines 67-69, is this a reference to the klotho knockout mouse?  If so then the statement should add the qualifier "in mice" and cite the paper or an appropriate review describing this mouse. If the sentence refers to human data on klotho expression and aging please include the specific citation and be more specific in this sentence about what was found.

Ad 4 The sentence refers to a mutant mouse that displayed phenotypes resembling human  premature aging syndromes therefore (additional reference has been added to clarify that Kuro-o Nature 1997, 390, 45-51)

5) Citation 11 (Ahmadi et al.) reported lower Klotho levels and similar FGF-23 levels in SSc patients compared to controls, and citations 12 (Hajialilo et al.), 13 (Talotta et al.), and 14 (Talotta et al.) found lower Klotho levels in SSc vs controls. The authors attribute the discrepancy in klotho levels to measurement of alpha-klotho levels (in this study) vs total klotho levels in the other studies.  Please comment on the distinct role of alpha-klotho compared to other subtypes, and why alpha-klotho was specifically measured in this study rather than all klotho.

Ad 5. The Klotho family consists of three single-pass transmembrane proteins—αKlotho, βKlotho and γKlotho. Each of them combines with FGF receptors (FGFRs) to form receptor complexes for various FGF’s. αKlotho is a co-receptor for physiological FGF23 signaling and appears essential for FGF23-mediated regulation of mineral metabolism and including phosphate homeostasis.

6) Page 3, lines 115-116: please include the specific catalog numbers or other identifying information for each ELISA kit in case the companies sell more than one for that target protein.

Ad 6: Catalogue numbers provided

7) Table 1: The mRSS distribution of 11.2 +/- 7.3 seems lower than expected for a cohort including only Diffuse patients.  This implies at least one patient had a score of only 4, correct?

Ad 7. It is entirely true. Two of our patients have mRSS value of 4. The subtype of disease was however established earlier therefore patients were categorized as having dSSc.

8) Table 1: I would change "Pulmonary hypertension" to "Elevated RVSP on echo" or "echocardiographic evidence of Pulmonary hypertension," since echo parameters were used rather than right heart cath.  It is ok and understandable that right heart caths weren't done, but it is misleading to say without qualification that these patients had pulmonary hypertension based on echo alone.

Ad 8. The pressure in right ventricle  has been measured indirectly I am entirety agree that description needs more clarification .”Elevated RSVP on echo” substituted “pulmonary hypertension” in table as requested  In the body of manuscript the sentence has been changed to the following To identify patients with echocardiographic evidence of pulmonary hypertension all patients underwent routine echocardiographic examination. Patients with estimated pulmonary arterial pressure on echocardiographic examination greater than 41 mm Hg were recognized as having elevated right ventricular systolic pressure (RVSP) on echo as recently proposed.

9) It would have been better to measure and report the creatinine values of the healthy controls in addition to the SSc patients,  since klotho levels have an association with GFR.

Ad 9. Unfortunately data on creatine level in controls are not actually available. We can do it having more than a week of time.

10) Page 5, line 148: Don't say "We failed to show any differences..." Just say "no differences were observed."

Ad 10. The sentence has been changed as proposed to: ‘No differences were observed when we analysed 25-OH-D levels in patients with scleroderma-related complications.’

11) page 5, lines 157-158: the p of 0.07 does not portray the strength or weakness of the correlation observed. Please provide the correlation co-efficient.

Ad 11. The sentence has been removed simply it was editing error. There was no correlation between ESR and Klotho (r = -0.19 p = 0.21). Additionally the revised manuscript has been double checked for potential similar errors.

12) page 5, Table 2: I don't understand why ESR was included in this table.

Ad 12: ESR removed from the table

13) page 5, lines 164-166: awkwardly worded. Can it just say "No associations were observed between …" and then list the variables that were analyzed?

Ad 13: The sentence has been corrected as proposed to: ‘In our group of patients no association was observed between vitamin D, α-Klotho level and iFGF23 on one side and extent of skin involvement (mRSS value), disease severity (Medgser scale) and activity (Eustar2017 scale) on the other.’

14) page 6, figure 1: please explain why these data were log transformed

Ad 14: ‘FGF/Klotho values were presented with logarithmic normalization due to skewed data distribution.’ This information was added at the bottom of Fig1.

15) page 6, figure 1: It would be informative to perform separate analyses with adjustments for age, smoking status, GFR, ESR, and vitamin D levels, i.e., other variables that might be associated with FGF23 and/or Klotho levels. Could the association of low FGF-23 with SSc have been driven in part by the association between low Vitamin D and SSc?

Ad 15: The suggestion of the reviewer is very valuable, however the sample size is too small to make so many adjustment. Because of that we omitted such analysis in the paper.

16) Please provide more detail of the disease activity metrics.  Were there a few that were particularly associated with the FGF23/Klotho index?

Ad 16: To give more insight the following sentence has been added: ‘Contrary to this none of disease activity metrics alone correlated with FGF23/Klotho index that included mRSS (r=0,08 p=0,55), total Medsger score (r=0,15 p=0,29). delta skin- (worsening/ improvement of skin status reported by patients), digital ulcers, tendon friction rubs, C- reactive protein and reduction of DLCO<70% of the predicted value).’

17) page 6, lines 181-182: I can't tell what is meant by "VD deficiency seems to be mostly related to disease per se."

Ad 17: I agree that is not precise statement therefore I changed it to: ‘Therefore, VD deficiency is not only related to sun exposure.’

18) page 6, lines 184-186: Another explanation of low vitamin D in SSc is impaired GI absorption.

Ad 18: Yes, indeed. the following sentence has been added: ‘Finally, majority of SSc patients suffer from GI tract involvement resulting in intestinal malabsorption that contribute to reduced vitamin D absorption.’

19) page 6-7, lines 203-215. I have a hard time understanding this discussion. In citation 11 the klotho levels were lower and FGF-23 levels similar in SSc.  Therefore the FGF-23/klotho index would have been in the opposite direction (higher nominator, lower denominator).  It might be better just to say that the reasons for the observed discrepancies are not yet clear.

Ad 19: Thank you very much for this comment. As proposed, substantial part of this paragraph has been removed,  accordingly. I remove ref 22 and summarized with sentence You proposed.

20) page 7, line 121: I recommend not writing "it is a very important observation." Leave that to the readers' judgement.

Ad 20: I agree one should not be the judge in his own case I changed word ‘important’ to ‘new’. Since at the moment the importance of this finding is not clear yet.

21) line 122: please change "previous studies failed to show any relationship." An observation of no association is not a failure if the study was done well. Please say instead that "a relationship was not observed" or something that doesn't imply a judgment of success or failure depending on whether association was found.

Ad 21: Sentence was changed as requested to: ‘It is a new observation as in the previous studies  any relationship between Klotho, iFGF23 and SSc activity was not observed.’

22) page 7, line 222-223, the findings here provide evidence of an association between FGF-23/Klotho and SSc activity. I'm not sure what is meant by "a direct link."  The association might be in either direction or without any cause/effect relationship at all, and might be influenced by countless other factors.

Ad 22: The sentence was changed to: ‘The observations from our study suggest association between  iFGF23/Klotho axis and activity of SSc, however this relationship is more complicated that previously suspected as it involves at least three elements….’

23) page 7, lines 227-228: what is the evidence that iFGF/Klotho index is a marker of disease activity "in states of VD deficiency?"  Was the relationship only observed in SSc patients with low vitamin D?  The authors didn't report the associations in subgroups of patients with normal vs low vitamin D.

Ad 23: Many thanks for this comment indeed relationship between normal vs low VD level in SSC was not tested. Therefore sentence was changed to: ‘Therefore we suggest that iFGF/Klotho index is a potential marker of diseases activity that may reflect diseases activity changes in SSc patients.’

24) page 8, line 231: please be more specific about "statistical bias." Do you mean that the results could have been influenced by random chance due to low sample size, or, that there were biases?

Ad 24: Some of analyses were not performed due to small sample ( see also p.15) Small sample was addressed in limitation paragraph and additionally I added the sentence to clarify that ‘however the relatively small group of subjects studied may have a strong impact on statistical analyses, final conclusions may therefore not be free of bias.’

25) In the methods section, please comment on whether the investigator measuring disease activity index was blinded to the lab variables (FGF-23, Klotho, vitamin D) under study.

Ad 25: To reduce interobserver variability and potential influence of lab results to clinical assessment  the clinical data and measurement of damage  activity indices has been collected by the same blinded assessor. The new sentence was added: ‘Disease characteristics as well as disease activity and damage assessment were performed by the same assessor who was blinded to results of Klotho FGF23 and VD.’

26) I think for work of this nature, looking for biomarkers of SSc disease activity, it should be discussed in the background or discussion what the current state of knowledge is on biomarkers in SSc.  Are there any other biomarkers with associations with SSc disease activity? If so what are they, and how strong were the associations?

Ad 26. At introduction I discussed the current state on biomarkers in SSC:

At the moment there are no unique  tools that may easily predicts the disease course and outcome and more importantly characterize patients prone to rapid disease progression. Currently patients’ clinical assessment relies on subjective measures  as modified Rodnan skin score (mRSS), or requires expensive or invasive procedures (e.g. heart catherization or HRCT) The diversity of disease is also challenging  as two distinct form of disease exist. Therefore there is a need for discovery of useful, easily available and cheap biomarkers that reflect fibrotic activity in the skin and  internal organs as well as ongoing vasculopathy. To date many potentially useful biomarkers have been proposed [2]. Majority of them are useful in prediction of involvement of a specified organ but do not reflect disease activity in the whole body[3]. Quite recently the European Scleroderma Trails and Research group (Eustar) proposed new revised activity  criteria for systemic sclerosis. Eustar index included such variables as patients skin assessment, extent of skin involvement measured by mRSS, tendon friction rubs, digital ulcers, DLCO value and C-reactive protein concentration [4]. Unfortunately Eustar index has not been tested against any currently available biomarkers yet

Reviewer 2 Report

The manuscript by Kotyla et al. describes the potential use of fibroblast growth factor to alpha-Klotho index as a marker for disease activity. Though these kinds of studies could help in better monitoring the disease activity of systemic sclerosis patients, there are certain limitation of this manuscript in its present form. For example:

1.     Materials and methods section is written in a complicated and confusing manner. I would recommend the authors to break the section into small subsections and then describe the procedure in a simplified way.

2.     The data provided is not strong enough to fully support the authors conclusion. More parameters showed be taken into considerations which are being directly co-related with fibroblast growth factor 23 or alpha-klotho before the index between two could be used to serve as a diagnostic marker.

3.     There are certain sentences in the manuscript that are confusing and difficult to understand. I would recommend the authors to change all those sentences in the manuscript.  Among all, few are listed below:

·       Line 102: In patients treated with cyclophosphamide assessment was performed at least 28 day after last infusion in order to reduce the drug related bias laboratory.

·       Line 122: Vitamin D status were classified on the base of 25-OH-D level as….

Author Response

We do really appreciate Reviewer attitude to improve our work. We did follow the suggestions and revised text subsequently.

 Reviewer 2

Materials and methods section is written in a complicated and confusing manner. I would recommend the authors to break the section into small subsections and then describe the procedure in a simplified way.

Ad 1: I divided the material& methods section into following paragraphs: inclusion and exclusion criteria, imaging and echocardiographic studies, concomitant medication

The data provided is not strong enough to fully support the authors conclusion. More parameters showed be taken into considerations which are being directly co-related with fibroblast growth factor 23 or alpha-klotho before the index between two could be used to serve as a diagnostic marker.

Ad 2: I agree that is only a proposal for potential marker and obviously importance of this finding should be verified, therefore some of limitation that may potentially contribute to reduction of data importance have been addressed in limitation of the study section in line with it several changes in the body of manuscript were done e.g The observations from our study suggest only association between  iFGF23/Klotho axis and activity of SSc, however this relationship is more complicated that previously suspected as it involves at least three elements: Klotho, VD and FGF23.

Therefore we suggest that iFGF/Klotho index is a potential marker of diseases activity that may reflect diseases activity changes in SSc patients. This finding however should be tested in larger study and more parameters should be taken into account as PTH, renal function and cytokine profile.

 There are certain sentences in the manuscript that are confusing and difficult to understand. I would recommend the authors to change all those sentences in the manuscript.  Among all, few are listed below:

I asked the friend ( senior lecture in Oxford University) for help in this issue. The manuscript has been intensively checked and corrected by the professional editing team provided by MDPI for grammar, spelling errors and rewritten in order to better understand

·       Line 102: In patients treated with cyclophosphamide assessment was performed at least 28 day after last infusion in order to reduce the drug related bias laboratory.

With the help of English native speaker we changed to the following . In patients treated with cyclophosphamide, the assessment was performed at least 28 days after the last infusion in order to minimize the direct influence of cyclophosphamide on parameters studied in the laboratory analysis.

·       Line 122: Vitamin D status were classified on the base of 25-OH-D level as….

Vitamin D status was classified on the basis of predefined 25-OH-D levels as: optimal >30mg/dl, insufficient - between 10 mg/dl and 30 mg/dl and<10mg/dl deficient as recently proposed

Round  2

Reviewer 1 Report

1) The writing and grammar quality is improved in this version, although a few minor errors remain.

2) Most of my comments about the first version have been well-addressed.

3) Regarding the potential for confounding, the authors said (in response to comment 15) that adjustments were omitted due to small sample size.  However I think that, at a minimum, an analysis should be done of the association between FGF23/Klotho index and SSc activity after adjustment for vitamin D levels.  The authors observed reduced vitamin D levels in SSc patients compared to controls, and this could impact FGF23 and/or Klotho levels.  I think it's appropriate to ask to what extent the association between FGF23/Klotho index and SSc activity was driven by low Vitamin D levels. Additionally, the authors report that no association was observed between vitamin D and disease activity; please report the correlation coefficient and P value for this analysis. Also please report the correlation coefficients and P values for FGF23 level and disease activity, and for Klotho and disease activity.

4) If the data distribution is skewed, is Pearson correlation the best statistical method?

Author Response

Thank you very much for your valuable comments. Please find the clarifications listed below:

Additionally, the authors report that no association was observed between vitamin D and disease activity; please report the correlation coefficient and P value for this analysis. Also please report the correlation coefficients and P values for FGF23 level and disease activity, and for Klotho and disease activity. 

The correlation coefficient for the association between vitamin D and iFGF23 was reported in the last version [There was a weak positive correlation between log10 levels of 25-OH-D and iFGF23 (r = 0.27; p < 0.05].  As suggested we show in the current revised version all the correlation coefficients and p values for FGF23 level and disease activity, and for Klotho and disease activity in the additional table – see below.

 Table 3. The Pearson correlation coefficient and significance level between Vitamin D, FGF 23, Klotho and FGF23/Klotho index and disease activity scales (Rodnan, EUSTAR, Medsger) in patients with SSc

Total EUSTAR [pts]

Medsger

mRSS

log10(iFGF-23 [pg/mL])

r = 0.23; p = 0.11

r = 0.02; p = 0.90

r = 0.09; p = 0.52

a-Klotho   [pg/mL]

r = -0.14; p = 0.33

r = -0.21; p = 0.14

r = 0.10; p = 0.48

log10(VD [ng/mL])

r = 0.02; p = 0.92

r = -0.01; p = 0.97

r = -0.02; p = 0.92

log10(iFGF-23/a-Klotho)

r = 0.35; p < 0.05

r = 0.15; p = 0.29

r = 0.09; p = 0.56

 If the data distribution is skewed, is Pearson correlation the best statistical method?

 All data were check whether the data distribution is normal or skewed. All necessary requirements were fulfilled for normal data distribution and in case of skewed distribution they were fulfilled after logarithmic normalization. Thus we could use the Pearson linear correlation coefficient. We added the information of logarithmic transformation into the paper.

 Regarding the potential for confounding, the authors said (in response to comment 15) that adjustments were omitted due to small sample size.  However I think that, at a minimum, an analysis should be done of the association between FGF23/Klotho index and SSc activity after adjustment for vitamin D levels.  The authors observed reduced vitamin D levels in SSc patients compared to controls, and this could impact FGF23 and/or Klotho levels.  I think it's appropriate to ask to what extent the association between FGF23/Klotho index and SSc activity was driven by low Vitamin D levels.

 The following explanation has been added to the body of manuscript

We have divided the group of patients with SSc into two subgroups: with vitamin D levels below (N = 37)  and over 30 ng/mL (N = 11). Based on this we have done covariance analysis with Total EUSTAR values and checked in if there is an influence of vitamin D concentration and disease activity on the log10 iFGF23/α-Klotho index. There was no statistically significant influence of vitamin D insufficiency/deficiency on the index (p = 0.97). However, a significant influence of Total EUSTAR was observed (p < 0.01), but without statistically significant interaction between these two analyzed variables (p = 0.15). Based on this we may conclude, that the correlation between the log10 iFGF23/α-Klotho index and disease activity is independent to vitamin D concentration.

Reviewer 2 Report

I thank the authors for giving a satisfactory response to all my queries.